# New Peptide-Based Pharmacophore Activates 20S Proteasome

**DOI:** 10.3390/molecules25061439

**Published:** 2020-03-22

**Authors:** Paweł A. Osmulski, Przemysław Karpowicz, Elżbieta Jankowska, Jonathan Bohmann, Andrew M. Pickering, Maria Gaczyńska

**Affiliations:** 1Department of Molecular Medicine, UT Health San Antonio, Texas, TX 78245, USA; ampickering@uabmc.edu; 2Barshop Institute for Longevity and Aging Studies, UT Health San Antonio, Texas, TX 78245, USA; 3Department of Organic Chemistry, Faculty of Chemistry, University of Gdansk, 80-308 Gdansk, Poland; przemyslaw.karpowicz@ug.edu.pl; 4Department of Biomedical Chemistry, Faculty of Chemistry, University of Gdansk, 80-308 Gdansk, Poland; elzbieta.jankowska@ug.edu.pl; 5Southwest Research Institute, San Antonio, Texas, TX 78238, USA; jonathan.bohmann@swri.org; 6The Glenn Biggs Institute for Alzheimer’s & Neurodegenerative Diseases, UT Health San Antonio, TX 78229, USA

**Keywords:** proteasome, activation, allostery, peptides, beta turn, proteasome dynamics, atomic force microscopy

## Abstract

The proteasome is a pivotal element of controlled proteolysis, responsible for the catabolic arm of proteostasis. By inducing apoptosis, small molecule inhibitors of proteasome peptidolytic activities are successfully utilized in treatment of blood cancers. However, the clinical potential of proteasome activation remains relatively unexplored. In this work, we introduce short TAT peptides derived from HIV-1 Tat protein and modified with synthetic turn-stabilizing residues as proteasome agonists. Molecular docking and biochemical studies point to the α1/α2 pocket of the core proteasome α ring as the binding site of TAT peptides. We postulate that the TATs’ pharmacophore consists of an N-terminal basic pocket-docking “activation anchor” connected via a β turn inducer to a C-terminal “specificity clamp” that binds on the proteasome α surface. By allosteric effects—including destabilization of the proteasomal gate—the compounds substantially augment activity of the core proteasome in vitro. Significantly, this activation is preserved in the lysates of cultured cells treated with the compounds. We propose that the proteasome-stimulating TAT pharmacophore provides an attractive lead for future clinical use.

## 1. Introduction

As the central protease of the ubiquitin–proteasome pathway, the proteasome has long been considered an attractive target for drugs potentially affecting multiple aspects of cell physiology [1]. Indeed, small molecules targeting the proteasome have entered the clinic with great success [2]. However, their scope at present is very limited: all proteasome-modifying compounds currently approved or clinically tested as drugs are competitive inhibitors and all are used to treat advanced blood cancers [1,3]. Here we turn to the opposite side of pharmacological intervention into the proteasome: augmentation of catalytic activity. Since dysfunction of proteasome-mediated controlled protein degradation is a hallmark of both cellular aging [4,5] and neurodegenerative diseases [6,7,8,9], enhancement of the enzyme’s activity should be considered an attractive therapeutic option. The complex structure of the catalytic core 20S proteasome (the “core particle”; Figure 1A) presents fascinating options for allostery-based augmentation, comprehensively reviewed in this issue [10,11]. The peptidase responsible for post-hydrophobic (chymotrypsin-like, ChT-L) cleavages is considered a rate-limiting “workhorse” and is the major target for inhibitors and activators alike [3,12]. Indeed, overexpression of a catalytic subunit harboring the active site of the ChT-L peptidase has been shown not only to extend lifespan but also to reduce age-related cognitive decline in animal models [13,14,15]. However, reports on pharmacological augmentation are limited to in vitro and cell culture studies. [16,17,18,19,20]. 

Here, we describe a series of short, modified peptides based on the basic domain of the viral Human Immunodeficiency Virus-1 (HIV-1) Transcriptional Activator TAR (Tat) protein (Figure 1B). Among many intracellular effects, the HIV-1 Tat protein inhibits the core proteasome [21]. In our previous studies, we noted that short peptide fragments of HIV-1 Tat displayed peculiar in vitro properties: they inhibited detergent-treated core particle but mildly activated the latent core [22,23,24]. However, treatment with sodium dodecyl sulfate detergent, although convenient for in vitro assays, yields mildly denatured proteasome and, under these far-from-physiological conditions, likely with destroyed natural allosteric routes [10]. Therefore, we turned our attention to the activating properties of HIV-1 Tat protein-derived “TAT peptides”. After observing a strong in vitro proteasome augmentation by modified HIV-1 Tat-derived peptides, we tested selected compounds in cell culture. In a separate study, we found that proteasome stimulation by TAT peptides partially prevented cognitive deficits and mortality in animal models of Alzheimer’s disease [15]. The very encouraging results included increased proteasome-mediated turnover of amyloid precursor protein (APP) and β-secretase (which cleaves APP to generate β-amyloid peptide), concomitant with lowered levels of β-amyloid, lowered mortality and protection against cognitive decline [15]. 

## 2. Results and Discussion

### 2.1. Design of TAT Peptides

We developed a set of proteasome agonists designed to activate the proteasome in vitro, to support blood-brain-barrier (BBB) transition, and ultimately to stably augment the proteasome in the nervous system. Our rational design was based on a basic domain of the Human Immunodeficiency Virus-1 (HIV-1) Transcriptional Activator TAR (Tat) protein ^48^GRKKRRQRRRPS^59^ (TAT1; (1)), which contains a proteasome-binding RTP (REG/Tat- proteasome-binding site) motif [26] This RTP motif is shared with subunits of the endogenous PA28/REG protein (proteasome activator/regulator with 28 kDa subunits; 11S), which targets pockets on the core proteasome α face [21] (Figure 1A). These pockets are established allosteric hotspots and specific binding sites not only for the REG activator but most importantly for the Rpt (Regulatory Particle ATP-ase) subunits of the 19S component of 26S proteasome. This holoenzyme is the most advanced and physiologically involved among assemblies sharing the 20S proteasome catalytic core [28]. Beyond their proteasome-binding capacity, TAT peptides have a strong cell-penetrating potential: a fragment nearly identical to TAT1, ^47^Y-R^57^, was previously reported as a “cell-penetrating-peptide” with high blood-brain barrier (BBB)-passing capacity due to its structure and highly positive charge [29]. Importantly, the basic domain extracted from the HIV-1 Tat protein context is devoid of either the transactivation or E3 recruitment capability of the viral factor [26,27,30] (Figure 1B). The whole Tat protein not only inhibits the core proteasome but also competes with the PA28/REG activator [21]. As mentioned above, we found before that TAT1 inhibits the artificially activated 20S core in vitro, but noted that it can activate the naturally latent core [22,23,24]. The goal of our design was to enhance the proteasome activation and improve the peptide’s stability while preserving BBB penetrance. This goal was accomplished by generating structural derivatives of the TAT1 peptide that contained turn-inducing moieties and the preserved key basic residues (Table 1). Our previous structural studies of the TAT1 peptide suggested a strong preference toward formation of two β-like turns in positions 4,5 and 8,9 [22,24] (Figure 1B,C). Substitutions of residues in TAT1 with alanines (Ala-walking) confirmed the significance of the putative turn regions for core proteasome activation (Figure 2). 

As the next step, we explored effects of introducing TO (Tic: L-1,2,3,4-tetrahydroisoquinoline -3-carboxylic acid, Oic: Octahydroindole-2-carboxylic acid), TOD (Tic D-Oic) or DABA (3,5-diaminobenzoic acid) as synthetic turn inducers in the critical positions of TAT1. Since the putative turn was flanked by predominantly basic sequences, we also attempted to explore the robustness of the design by simplifying the Arg and Lys stretches to triple-Lys (Table 1).

### 2.2. TAT Peptides Activate the ChT-L Peptidase of the Human Proteasome in vitro and in Cellulo

As summarized in Table 2 and demonstrated in Figure 3, TAT1 and peptides with turn inducers in either of the predicted turn-promoting positions activated the latent human core 20S proteasome. Stabilization of a turn in the position 8,9 seemed superior to the position 4,5 for activating potential. Importantly, simultaneous stabilization of both turns in (2) was detrimental to the activation capability. On the other hand, replacement of TAT1 basic fragments with a KKK sequence yielded (6) with activation preserved, albeit with disappointingly high AC_50_. The negative control (7), with an AAA stretch replacing KKK, displayed no ability to activate the core proteasome. With the goal of improving affinity of (6) to the proteasome, we supplemented it with an additional KKK sequence and joined all three triple-Lys fragments via diaminobenzoic acid in (8) (Tat1-Den). Consequently, (8) displayed one of the highest activation folds for the 20S core and one of the lowest AC_50_ (Table 2, Figure 3). 

Interestingly, we observed two types of titration profile with TAT peptides (Figure 3): The most common was a profile with a pronounced maximum of activation followed by a systematic drop of the effect, resulting in a bell-shaped curve. We suspect that this type of response is characteristic for compounds that have additional binding sites with a weaker apparent binding constant and negative cooperativity with the primary binding site. The role of the secondary binding site can be played by the homologous sites present on the opposite side of 20S complex. The primary binding of these TAT ligands may stimulate secondary binding sites characterized by distinct thermodynamic properties. The secondary sites may bind accumulating products of substrate digest instead of TAT ligands and trigger partial inhibition of the activation effect. Finally, binding of the ligands may affect not only performance of a single active center but cooperativity between them. It is established that the active centers responsible for breaking peptide bonds at hydrophobic residues (ChT-L activity) are linked by positive cooperativity [31,32]. The second type of response followed a typical sigmoidal shape reaching the saturation level. This titration profile was found in only three cases: a very poor activator (2) containing two turn inducers, a good activator (6) with one TO turn and two simplified KKK “whiskers”, and a similar but DABA-based dendritic peptide with three KKK “whiskers” (8). The most straightforward explanation would imply that, for these compounds, any meaningful secondary site binding and accompanied activity drop required impractical or unachievably high ligand concentrations. Alternatively, we may speculate that these compounds simply do not have secondary binding sites that induce proteasome inhibition. Instead, they may have additional binding sites that prompt the positive cooperativity. The current biochemical data preclude conclusion if this titration shape reflects binding only to the analogous site on the other side of the proteasome molecule or additional unidentified binding pockets.

Among the three proteasomal cleavage specificities only the “workhorse” ChT-L peptidase was significantly affected, as demonstrated for selected compounds (Figure 2 and Figure 3). Compounds (5) and (8) actually showed weak inhibitory effects on T-L (trypsin-like) and postglutamyl peptide hydrolyzing (PGPH) activities (Figure 4). 

Taken together, a structural constraint induced by a single turn placed close to the C-terminus and selected basic residues were necessary to achieve the strongest activation at the lowest peptide concentration. 

The strong in vitro performance of TAT peptides inspired us to test the influence of selected activators on proteasome activity in cultured cells. We chose the human neuroblastoma SK-N-SH (ATCC/American Type Culture Collection HTB-11) line as representative for neural cells as these may become future targets of proteasome agonists in treatment of neurodegenerative diseases with compromised proteasome performance. None of the tested compounds at 1 µM significantly affected proliferation and viability of the cells after 24 h of treatment (see caption of Figure 5). However, the proteasome activity in lysates prepared from the treated cells was significantly higher than activity in control lysates for (5) and (8). The lack of significant activation for (1) (Figure 5, left) could be explained by significantly compromised cytosolic stability of the TAT1 peptide, as compared with the derivatives that may be protected from degradation by the synthetic turn inducers. This conclusion is further supported by the observation that dose dependent proteasome activation with (5) in SK-N-SH cells was substantially stronger after 4 h exposure than after 24 h (Figure 5, right). Importantly, both (2) and (9) followed in cellulo their poor in vitro performance. The activation by (5) and (8) detectable in the lysate could be explained by two non-excluding phenomena: the direct enhancement of the peptidolytic activity by strongly binding compounds or enrichment of the content of active proteasomes in treated cells. Our pilot experiments point at the strong binding of these otherwise reversible compounds. Since 1 µM bortezomib treatment of the lysates from both control and TAT-exposed cells abolished ChT-L activity to a comparable extent (>85% (Figure 5)), we excluded the possibility that the observed increase of peptidase activity was due to upregulation of other proteases. Future studies will test the possibility of further effects on the level of expression of proteasome subunits and shifts in partition of distinct proteasome assemblies.

### 2.3. The α1/α2 Inter-Subunit Pocket on the α Face of Core Proteasome is the Primary Binding Site of TAT Peptides. The Binding Changes Conformational Equilibrium of the Proteasome’s Gate

To gain mechanistic insight and to aid further modifications of the compounds, we performed molecular docking of (5) to the human core proteasome using Rhodium^®^ software suite (Southwest Research Institute; SwRI; San Antonio). In its docking approach, different locations on the surface were seeded with 10^4^ to 10^5^ copies of a ligand conformer, generating trial candidate binding configurations. The inhibitor’s seeded configurations were allowed to move independently over the surface, optimizing the coordinates of the binding location along a path to a local energy minimum on the surface. Certain ligand molecules starting at different locations converged to several common locations. The docking was performed in two steps. First, a square-well interatomic potential for docking, similar to the approach published by Vakser [33,34] was used for the primary docking. Next, the identified docking pose candidates were screened with a second tier docking for pose refinement, typical for the traditional docking codes.

Figure 6A demonstrates the proposed docking with (5) positioned in the pocket between the core subunits α1 and α2. The N-terminal part of (5) was proposed to penetrate deep into the pocket, with multiple potential hydrogen bonds engaging all the N-terminal (1−7) residues of (5) and stabilizing the binding. The predicted turn extended above the α face, whereas the C-terminal Ser showed propensity to interact with exposed residues of α1 (Figure 6B). Based on this result, we propose that the N-terminal part of (5) may serve as an “activation anchor”, whereas the turn positions the C-terminal part to interact with the α face as a “specificity clamp” (Figure 6C). The improved performance of (8) as compared with (6) could be then explained by beneficial actions of an additional “specificity clamp”, possibly strengthening the ligand binding (Figure 6D). 

As mentioned above, the binding pockets on the α face accept “anchors” from natural protein ligands of the 20S core [25,36,37]. Certain anchoring peptides are known to interact with the α face in trans, most notably C-terminal “tails” of Rpt ATPase subunits of the 19S complex bearing the “HbYX” (hydrophobic-Y-any amino acid) C-terminal motif [38]. The 10-residue Rpt-derived C-terminal “tail” peptides (“Rpt peptides”) can be used as competitors with the Rpt subunits or with small allosteric ligands [19,38,39]. Importantly, the Rpt peptides interject between the subunits with their C-termini, whereas TAT peptides, according to the modeling, use their N-termini for this purpose. To test the specificity of interactions between TAT compounds and the α face, we performed competition experiments. The 20S core was challenged with selected TAT compounds after treatment with Rpt peptides. We selected peptides of Rpt2 (QEGTPEGLYL), Rpt3 (KDEQEHEFYK), Rpt5 (KKKANLQYYA) and Rpt6 KNMSIKKLWK) subunits, docking in α3/α4, α1/α2, α5/α6 and α2/α3 pockets, respectively. Tails of Rpt2, 3, and 5 display the canonical HbYX motif, whereas the LeuTrpLys C-terminus of Rpt6 may be considered “pseudo-HbYX”, with a bulky Trp replacing Tyr. Results of the competition experiments are presented as radar plots in Figure 7. Relative ChT-L proteasome activity at or near 1 indicated that activations by a Tat compound and by the Rpt tail (if detectable) were fully preserved and therefore no competition between the peptides was considered. A score below 1 indicated active competition, whereas a score above 1 suggested enhanced, possibly synergistic, activation (positive cooperativity) in the presence of a TAT and an Rpt tail. The indications of the molecular docking (Figure 6) were fully confirmed for (5): the Rpt3 peptide (α1/α2 pocket) was its sole competitor. This pocket also emerged as the major binding site for (1) and (8). However, these compounds likely weakly competed with the Rpt2 tail for binding to the α3/α4 pocket as well. The α3/α4 pocket seemed to be favored by (6). The poor activator (2) was, not surprisingly, least specific in its binding preferences, adding the spacious α5/α6 (Rpt5) pocket to the putative sites of competition. Interestingly, (1), (5), (6), and (8) displayed enhanced activation in the presence of one of Rpt tails, namely Rpt5 or Rpt6. The presumed synergy was especially pronounced for the Rpt6-(2) pair, as well as Rpt5-(1), (5), and (6) pairs (Figure 7). These complex patterns point at the importance of allosteric effects between the α face ligands, the gate and the catalytic chamber.

The putative binding site corresponds to one of the natural “anchoring spots” on the α face of the core proteasome. The inter-subunit pockets are used to attach regulatory proteins: PA28/REG (all pockets), PA200 (proteasome activator of 200 kDa; α5/α6 pocket), as well as the Rpt subunits of the 19S particle (all pockets except α6/α7 and α7/α1) [26,36,40]. Peptide-activators of the core that utilize the structure of docking fragments of these natural activators were found to bind into the α5/α6 pocket, while a small molecule activator TCH-165 reportedly preferred the α1/α2 site [19,20,41]. Binding of all these ligands resulted in opening or at least destabilizing the gate in the center of the α face, as revealed by crystal structures, cryoEM (cryo-electron microscopy) and atomic force microscopy (AFM) imaging [19,20,25,40,41,42,43]. Gate opening is prerequisite for the uptake of substrates and release of products from the concealed catalytic chamber of the core proteasome (Figure 1A). Conformational diversity of the latent core proteasome allows for periodic gate opening and potential substrate uptake [44]. Moreover, the gate is allosterically connected by a positive feedback loop(s) with active sites: a catalytic act in any of the sites sends a signal to open the gate and upkeep enzymatic action [45]. As we have established previously, AFM collects images of single, native proteasome particles at nanometer resolution. The particles exhibit a well discernible gate area, assuming dynamic “open”, closed” or “intermediate” positions, corresponding to the gate conformations detected by cryoEM [20,25,39,46]. These observations prompted us to search with AFM for structural consequences of (5) binding to the proteasome. The latent core is known to preferentially remain in the closed-gate conformation, accounting for about three-quarters of particles detected by cryoEM or AFM, with less than 10% assuming the fully open-gate position [20,25,39]. In contrast, treatment with 1 μM of (5) resulted in a dramatic shift in partition of conformations toward more abundant, over 40%, open-gate proteasomes, and less prominent, below 30%, closed-gate particles (Figure 8). Also, slightly (albeit significantly) more particles assumed the intermediate gate position in (5)-treated preparation than in vehicle-treated control (Figure 8). In control and compound-treated preparations alike, all single particles could freely change their conformations with time, as detected in consecutive scans. The conformational landscape with such a prominent representation of open-gate particles is unique among proteasome partitions detected in the presence of other small proteasome ligands. Pro-Arg-rich and HbYX-accommodating peptide activator PR2 prompted a strong increase in intermediate (nearly 40%) and only a moderate increase (less than 20%) in open-gate forms [20]. In contrast, allosteric small-molecule or peptide inhibitors docking in the α face pockets suppressed the open-gate conformers [20,39]. The apparent strong stabilization of the open-gate conformation by (5) would be expected to promote catalysis by easing the rate-limiting obstacle of a closed gate. However, the fact that only ChT-L and not the other two proteasomal peptidases are activated by TAT peptides (Figure 3 and Figure 4) also points at the possible role of allosteric signaling. Indeed, proteins, peptides or small molecules targeting the α face may in vitro affect one or more peptidases [28]. Explanation of the diverse effects may involve a direct signaling between the α face pockets and catalytic chamber, between the gate and catalytic chamber, or all the former plus inter-catalytic sites allosteric loops [47,48]. TAT peptides may join other allosteric ligands in providing powerful tools for studying proteasome allostery.

In the light of significant proteasome-enhancing effects of TAT peptides observed both in cellulo (Figure 5 and [15]) and in vivo [15], there is a crucial question of how they influence proteolytic activity in the most widespread form of the proteasome, the 26S holoenzyme. The most-explored TAT compounds are proposed to dock in the α1/α2 pocket, which, alongside the α5/α6 pocket, is permanently occupied in all conformational forms of the 26S assembly [46]. Therefore, activation of the core already decorated with two 19S modules seems excluded. Still, in our in vitro experiments, a significant, up to three-fold activation of the 26S proteasome was observed. Although it was lower than for the 20S proteasome, it was concentration-dependent ([15] and Osmulski, Gaczynska, *unpublished observations*). The two most plausible explanations of the phenomenon are: (i) the compounds affect only the free core present in small quantities in purified 26S preparations and present to a physiologically determined extent in living cells; (ii) the previous case extended with activation of the half-26S assemblies with only one α face blocked by 19S cap and the other left free. Since single-cap proteasomes are fully capable of recognizing and processing polyubiquitinated substrates, the latter opens an intriguing opportunity of TAT peptides mediated activation of ubiquitin-dependent proteolysis. Importantly, as revealed by cryoelectron tomography in live neurons, two-thirds of 19S-decorated proteasomes are in the single-capped form [49], whereas in many other cells, full 26S assemblies or hybrids with both α faces blocked were reported as most abundant [50,51]. Such opportunity to efficiently activate proteasomes in neural cells would be especially relevant to potential anti-Alzheimer’s disease actions of the peptides. As reported by Chocron et al. [15], the proteins with turnover increased by treatment with (5) or (8) included proteasome substrates possibly processed in ubiquitin-independent (APP; [52]) but also in the ubiquitin-dependent manner (β secretase; [53]). Unraveling the physiological mechanism of action of TAT peptides as well as their drug-oriented optimization needs to precede their use as anti-Alzheimer’s disease therapeutics. 

## 3. Materials and Methods 

### 3.1. Synthesis of Selected Peptides

Synthesis and properties (1) have been described in [22], (2)–(5) and (9) have been described in [24]. Synthesis and purification of (6) and (7) followed the procedures described in [24]. All the peptides have been purified to at least 99% of purity. 

### 3.2. Synthesis of TAT1-Den Peptide (8)

Synthesis of (8) was performed on 0.25 mmol scale, according to Fmoc/tBu methodology, in a Liberty BlueTM automated microwave synthesizer (CEM Corporation). The TentaGel PHB resin was used as a solid support with an initial capacity of 0.23 mmol/g.

The following Fmoc-protected amino acid derivatives were used in the synthesis: Fmoc-Lys(Boc)-OH and Di-Fmoc-3,5-diaminobenzoic acid.

The first amino acid, Fmoc-Lys(Boc)-OH, was attached to the solid support using 1-methylimidazole (MeIm) and 1-(2-mesitylenesulfonyl)-3-nitro-1*H*-1,2,4-triazole (MSNT). 5 eq. of Fmoc-Ser(t-Bu)-OH (relative to the resin capacity) was dissolved in dichloromethane (DCM) with addition of a few drops of tetrahydrofuran. Next, 3.37 eq. of Melm and 5 eq. of MSNT were added, and the mixture stirred for 15 min. The mixture was then transferred to a round-bottom flask containing the resin swollen in DCM. The mixture was flushed with argon and left on a vertical shaker for 2 h, then the peptidyl resin was drained, washed and dried in a vacuum desiccator. The resin loading was determined as follows: a few milligrams of the dry peptidyl resin were transferred to a 2 mL test tube, 1 mL of 20% piperidine in dimethylformamide (DMF) was added, and the tube was shaken for 30 min. Then, the mixture was transferred to a 25 mL volumetric flask and filled with methanol. The solution was transferred to a quartz cuvette and the loading of the peptidyl resin was calculated from measurement of the absorbance at *λ* = 301 nm.

The Fmoc-Lys(Boc)-resin was transferred into a reaction vessel and soaked prior the synthesis cycle in DMF for 30 min. In the next step, the Fmoc group was removed (deprotection cycle) using 30% solution of piperidine in DMF. The mixture was irradiated for 15 s with a 167 W microwave power (temperature 75 °C), then with a power of 31 W for 50 s (temperature in the range 89–90 °C). The solid support was then drained and washed four times with DMF, and the deprotection cycle was repeated. Next N-terminally protected amino acid was attached, using as a coupling solution a mixture of 0.5 M *N,N*′-diisopropylcarbodiimide (DIC) and 1 M Oxyma pure (racemization suppressor) in DMF. The coupling reaction step was carried out with a four-fold excess of an amino acid derivative, calculated based on the initial capacity of the solid support. The efficiency of this step was enhanced with microwave radiation of 162 W for 15 s (temperature 75 °C), then 33 W for 110 s (temperature in the range of 89–90 °C). The peptidyl resin was then drained and the coupling cycle was repeated. Next, the 30% piperidine solution in DMF was added to de-protect the N-terminal amino group. This step was carried out in the same conditions as described above. Double coupling cycles with the use of DIC/Oxyma reagents were performed till the attachment of the third Fmoc-Lys(Boc)-OH residue (fifth residue in the sequence). Coupling of the fourth residue in the sequence (Di-Fmoc-3,5-diaminobenzoic acid) was carried out with a three-fold excess of the amino acid, calculated based on the initial capacity of the solid support. The efficiency of this step was enhanced by applying microwave irradiation (85 W for 60 s, temperature 40 °C, then 25 W for 540 s, temperature in the range of 63–65 °C). The peptidyl resin was then drained and the coupling cycle was repeated. The residue was deprotected under the same conditions as described above, with triple repetition of the cycle. Starting from the third residue in the sequence, the coupling reagents were switched to 1-Cyano-2-ethoxy-2-oxoethylidenaminooxy) dimethylamino- morpholino- carbenium hexafluoro phosphate COMU. The N-protected amino acid derivatives were coupled with the use of a three-fold excess of an amino acid, calculated based on the initial capacity of the solid support, 2.9-fold of COMU and 5.8-fold of diisopropylethylamine. The coupling efficiency was enhanced by applying microwave irradiation of 120 W for 60 s (temperature 60 °C), then 25 W for 30 s (temperature in the range of 78–80 °C). The peptidyl resin was then drained and the coupling cycle was repeated. The deprotection reagents and protocols were not changed. After completion of the synthesis, the peptidyl resin was washed four times with DMF, then three times with methanol and left overnight to dry in a vacuum desiccator.

#### 3.2.1. Peptide Cleavage from the Solid Support

The peptide was cleaved from the solid support, along with the removal of protecting groups from amino acid side chains, using the mixture of trifluoroacetic acid (TFA), triisopropylsilane and water (92:4:4, v/v/v). The reaction was carried out for two hours on a laboratory shaker. The resin was then drained under the reduced pressure on a filter funnel and the filtrate was concentrated to a volume of about 2 mL with a vacuum evaporator. The remaining filtrate was treated with diethyl ether (cooled to about 4 °C). A white precipitate was obtained and centrifuged in a centrifuge tube for 15 min (4500× *g*). The supernatant was decanted and the pellet was treated with another portion of diethyl ether. The precipitate was washed this way three times, and then dried in a vacuum desiccator. The obtained crude product was dissolved in water and freeze-dried.

#### 3.2.2. Purification

The compound was purified using a reversed-phase HPLC (RP-HPLC). The crude product was dissolved in water and injected onto a Jupiter^®^ Proteo C12 semipreparative column (21.2 mm × 250 mm, 90 Å, 4 μm; Phenomenex). The chromatographic separation was carried out using a linear gradient of 1-100% B over 75 min, and the eluents: A: 0.1% TFA in H_2_O and B: 0.1% TFA, 10% methanol in H_2_O. The eluent flow rate was 15 mL/min, UV detection at *λ* = 223 nm. After the collection of the main fraction, solvents were evaporated using a vacuum evaporator. Next, the fraction was dissolved in water and injected onto the same semipreparative column. The second purification was carried out in a linear gradient of 1–40% B over 75 min with the eluents: A: 0.1% TFA in H_2_O and B: 0.1% TFA in 5% acetonitrile (ACN) in H_2_O. The eluent flow rate was 15 mL/min, UV detection at *λ* = 223 nm.

#### 3.2.3. Characterization of the Product with HPLC and Mass Spectrometry

The product was subjected to chromatographic analysis using RP-HPLC. Conditions: chromatographic column: Kinetex 2.1 mm × 100 mm, 100 Å, 2.6 μm (Phenomenex); eluents: A: 0.1% TFA in H2O, B: 0.1% TFA, 80% ACN/H_2_O; flow rate 0.5 mL/min; UV detection at *λ* = 223 nm; gradient 5–45% B over 7 min, temperature of an oven 40 °C, Rt = 4.42 min. The calculated molecular weight of the compound was confirmed using a LC-MS IT-TOF (Shimadzu) mass spectrometer. Peptide was injected directly into the ion source. Theoretical average molecular weight of the compound: 1305.2 Da, obtained m/z: 1304.81 [M]+. 

### 3.3. Determination of Proteasome Activity

Human housekeeping core (20S) proteasome purified from erythrocytes was purchased from Boston Biochem, Inc. (Cambridge, MA) or from Enzo Life Sciences, Inc. (Farmingdale, NY). Multiple batches of the proteasomes were used and performed reproducibly. The stock proteasome was diluted to 0.2 mg/mL working solution in “dilution buffer” (50 mM Tris/HCl, pH 8, 20% glycerol). The following model peptide substrates, all releasing fluorescent 7-amino-4-methylcoumarin (AMC) reporter group after cleavage, were used: succinyl-LeuLeuValTyr-AMC (for the ChT-L peptidase; SucLLVY-AMC; Bachem Bioscience Inc., Philadelphia, PA), butoxycarbonyl-LeuArgArg-AMC (T-L; Bachem Bioscience Inc., Philadelphia, PA) and carbobenzoxy- LeuLeuGlu-AMC (PGPH; Enzo Life Sciences Inc., Farmingdale, NY). The substrates were used at concentration of 50 μM (ChT-L) or 100 μM (T-L, PGPH). Free AMC (Sigma-Aldrich, St. Louis, MO) was used as the standard. The C-terminal peptides derived from Rpt2, Rpt3, Rpt5 and Rpt6 were synthesized (standard solid-phase peptide chemistry) and purified to at least 98% purity by GenScript (Piscataway, NJ). The TAT peptides, Rpt peptides (except Rpt6) and the peptide substrates were dissolved in anhydrous dimethylsulfoxide (DMSO; Sigma-Aldrich, St. Louis, MO) and such stock solutions were stored at −20 °C. The total concentration of DMSO in final reaction mixtures never exceeded 3% (vol/vol). The Trp (tryptophan) -containing Rpt6 peptide was dissolved in ultrapure water and stored at −20 °C, protected from light. The reaction was carried out in 96-well plates, in 100 μL of reaction mixture that consisted of 45 mM Tris/HCl, pH 8, 100 mM KCl, 1 mM EDTA (reaction buffer) and a fluorogenic peptide substrate, to which 200 ng (nearly 0.3 nmol) of proteasome were added. The addition of KCl/EDTA (ethylenediaminetetraacetic acid) to reaction buffer assured latency of the core proteasome. The proteasome was preincubated with a substrate for 10 min at room temperature, then 1 μL of DMSO or a desired concentration of a TAT peptide in 1 μL of DMSO were added. After mixing, the plate was transferred to a Fluoroskan Ascent plate reader (Thermo Fisher Scientific Inc., Waltham, MA) for 1 h (37 °C), with fluorescence readouts once per minute [54]. To test the competition with Rpt derived peptides, the Rpt peptide (1 μM) was added before the TAT peptide to the reaction mixture. The reaction rates were calculated from a linear segment of kinetic curves constructed from measurements in 1-min intervals. Reaction rates were calculated using a linear fit performed with the Slope Analyser and Enzyme Kinetics applications launched within Origin Pro 2019 (OriginLab Corporation, Northampton, MA). Specific ChT-L activity of the latent control 20S proteasome ranged from 3.4 to 6.0 nanomoles of AMC product released by 1 mg of 20S per minute (4.2 ± 0.8; *n* = 26). All data are presented as mean ± SD from at least three independent experiments.

### 3.4. Cell Culture

Human SK-N-SH neuroblastoma cell line (ATCC^®^ HTB-11™American Type Culture Collection; Manassas, VA) were cultured according to ATCC specifications (EMEM, 10% heat-inactivated FBS) The cells at passage 2-4 were treated with 1 μM TAT peptides or the DMSO vehicle diluted with the medium 1: 1000, for 24 h. The content of live cells was determined by the Trypan Blue-exclusion assay. The cells were harvested, washed twice in PBS, resuspended in dilution buffer (see above) and stored in −80 °C. To prepare lysates the thawed preparations were vortexed with glass beads and centrifuged for 5 min 5000× *g* (4 °C). The supernatant was centrifuged for 20 min 14,000× *g* (4 °C). The resulting supernatant—“crude lysate” was diluted to 1 mg/mL of total protein with dilution buffer. 1 μg of lysate per assay (reaction buffer: 50 mM Tris/HCl pH 8, 0.1 mM MgCl_2_, 0.2 mM ATP, 0.1 mM dithiothreitol, 50 μM Suc-LLVY-AMC) was used for determination of ChT-L activity in a 96-well format, as above. Activities in lysates were also tested in the presence of a high concentration (1 μM) of a strong competitive proteasome inhibitor Bortezomib. The resulting negligible degradation of the model substrate in the presence of bortezomib indicated that proteasome is the sole source of activity observed in the lysates. 

### 3.5. Molecular Docking

Relative binding locations of (5) were determined on the surface of human core proteasome represented by crystal structure 5LE5 [35]. The previously reported 3D structures of (5) were used [24]. Docking poses were determined with Rhodium^®^ 3.9 in four separate docking trials, as described in Results. For each trial, poses were generated on a grid covering the surface of the protein model, with 72 trial states per grid point, with resolution of 1.7 Å. Forty poses with the maximum cavity-filling scores were prepared and analyzed with PyMol v.1.5.0.5 (Schrödinger LLC, New York, NY [55,56,57]). The top-ranking pose with ligand-proteasome contacts along both activation anchor and specificity clamp is presented.

### 3.6. Atomic Force Microscopy (AFM) Imaging

We used our established procedure to image single molecules of 20S proteasomes in tapping (oscillating) mode in liquid, with a scanner E of the Multimode Nanoscope IIIa (Bruker Inc., Santa Barbara, CA) [45]. The proteasomes were electrostatically attached to a muscovite mica substrate, covered with imaging buffer (5 mM Tris/HCl, pH 7) and scanned using cantilevers with the spring constant of 0.35 N/m from the SNL (Sharp Nitride Lever) probes (Bruker Inc., Santa Barbara, CA) tuned to 9–10 kHz. The amplitude set-point in the range of 1.5–2.0V, drive voltage of 300–500 mV and 3.05 Hz scanning rate were used. Scans of 1 µm^2^ fields (512 × 512 pixels) were collected in height-mode. The images of fields typically contained several dozens of top-view (“standing”) 20S proteasome particles. The gate status was deduced from a profile of raw height values of pixels measured by a probe scanning across the single proteasome particles, as previously described [20,39]. In short, under the employed scanning conditions the proteasome α face with the central gate area was completely rendered by a six-pixel (11–12 nm) scan-line fragment. Numerical values of the height of particles of raw (after standard flattening) images were collected with a practical vertical resolution reaching 1 Å. When this scan-line presented a local minimum (a central dip), the particle was classified as containing the open gate. The particle was classified as an intermediate conformer when a plot of height values presented a concave function without a local minimum. When the function was convex, the particle was classified as containing the closed gate. The “events” of gate opening/closing were analysed for scans of distinct particles as well as multiple scans of the same particles. 

### 3.7. Statistical Analysis

All experiments were performed at least in three independent replicates. The results are presented as a mean ± SD. A two-tailed t-test was applied to compare the means. The comparisons were allowing for unequal variances with a Welch correction. Abundance of proteasome conformers was compared with the chi squared test and two-sample proportion test. A significance level was set at 0.05. Statistical analysis was performed using statistical procedures offered by OriginPro 2019 (OriginLabs, Northampton, MA). Reaction rates were calculated from a smoothed linear segment of kinetic traces using OriginPro 2019. Response curves (activity vs. compound concentration) were fitted with a nonlinear fitting application of OriginPro 2019. The AC_50_ and maximum activation fold was calculated based on the equations implemented to fit the response curves.

## 4. Patents

The data presented above are included in the U.S. Patent Application no. 62/900,217 “Peptide-Based Compositions And Methods For Treating Alzheimer’s Disease”.

## Figures and Tables

**Figure 1 molecules-25-01439-f001:**
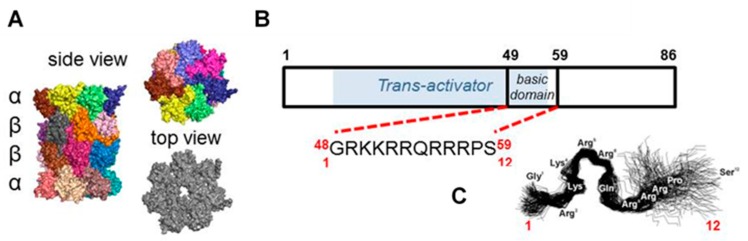
The human 20S core proteasome, Human Immunodeficiency Virus-1 (HIV-1) Transcriptional Activator TAR (Tat) protein and TAT1 peptide. (**A**) The crystal structure based model of the 20S catalytic core proteasome. Colored subunits of the tube-shaped core are arranged in four heptameric rings (αββα) that form the inner catalytic chamber and the outer “α face” equipped with a central gate for substrate uptake and product release. The pockets between α subunits are utilized to anchor 19S or other regulatory modules of the catalytic core. Based on Protein Data Bank structure pdb 5t0g [25]. (**B**) Scheme of HIV-1 Tat protein, with the domain required for transactivation [26,27] shaded blue and with the short fragment designated as TAT1 marked. (**C**) Our previously reported molecular dynamics simulation of TAT1, suggesting a potential for formation of two turns (modified from [22,24].

**Figure 2 molecules-25-01439-f002:**
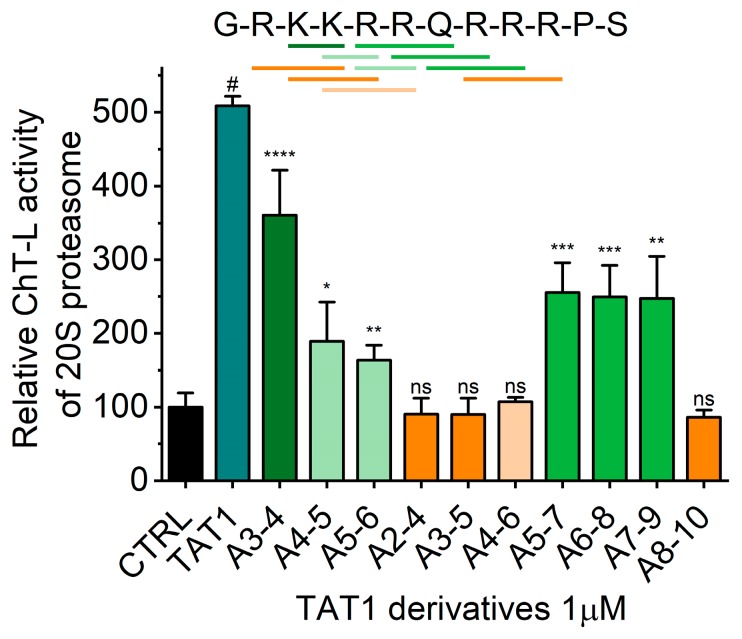
TAT1 derived peptides (1 μM) with selected residues replaced by alanine were tested for their activating capabilities of the chymotrypsin-like (ChT-L) activity of human latent 20S proteasome. Activity of 20S proteasome in the absence of any TAT peptide (DMSO) was set at 100%. Mean and SD of three independent experiments are presented. *Top:* The amino acid sequence of TAT1 peptide (1). Bars mark residues replaced by alanine. Green-like colors indicate no significant effect of the substitution, whereas orange-like colors mark hot spots resulting in loss of activation effect. Substitution of any single residue with alanine did not have a noticeable systematic effect on ChT-L activity. Columns show average + SD; *, *p* < 0.05; **, *p* < 0.005; ***, *p* < 0.0005; ****, *p* < 0.0001; #, *p* < 1 × 10^−7^; ns: not significant; *n* = 3–5, t-test performed against the activity of control, untreated 20S proteasome.

**Figure 3 molecules-25-01439-f003:**
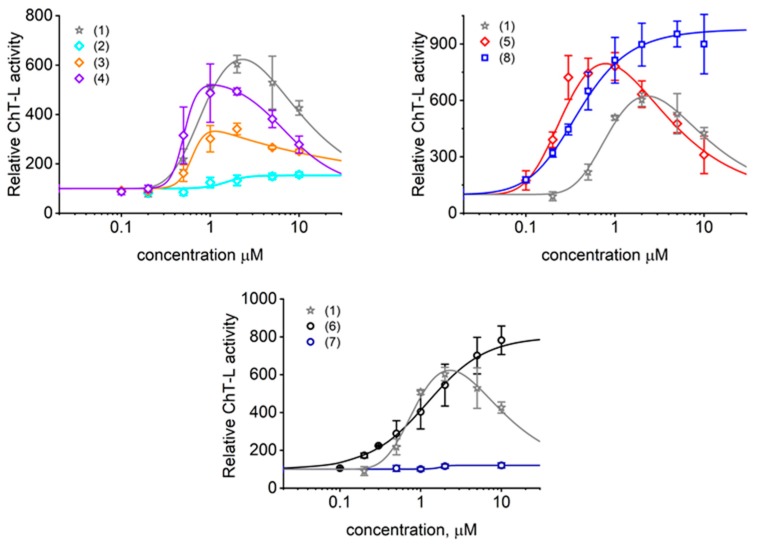
The chymotrypsin-like activity of latent human 20S proteasome was activated by TAT1 (1) and its derivatives in a dose dependent manner. (4)−(5) with turn inducers at the 8,9 position all outperformed (3) with a TO turn inducer at the position 4,5. Interestingly, (2) that contains TO and TOD turn inducers at the positions 4,5 and 8,9, respectively, was a very poor activator. (8) exhibited the highest fold of activation potential and a very low AC_50_. No ability to activate was detected for (7) where all lysines were substituted with alanines and for (9) with alanines replacing turn-promoting residues (not shown). Means ± SD, *n* = 3–8. The control activity presented as 100% corresponded to 5 nanomoles of the fluorescent AMC product released per minute by 1 mg of 20S proteasome from the Suc-LLVY-AMC substrate (succinyl-LeuLeuValTyr-7-amino-4-methylcoumarin).

**Figure 4 molecules-25-01439-f004:**
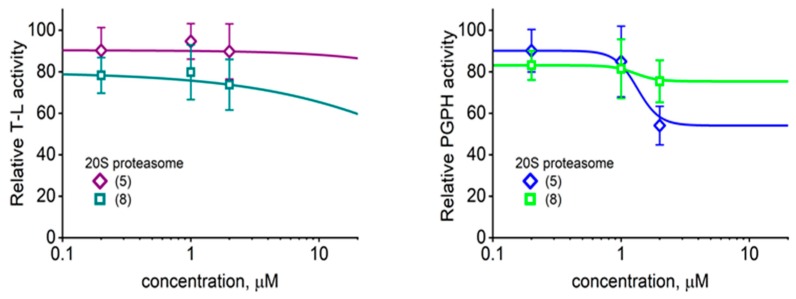
As demonstrated for (5) and (8), the TAT1 derivatives did not seem to significantly affect the post basic (trypsin-like; T-L) and post acidic (post glutamyl peptide hydrolyzing; PGPH) peptidase activities of the core proteasome. Means ± SD, *n* = 3–4. The control relative activities (100%) corresponded to 9.2 ± 1.4; *n* = 3 (T-L) and 3.1 ± 0.3; *n* = 4 (PGPH) nanomoles of the AMC (7-amino-4-methylcoumarin) product per mg of latent 20S per minute, respectively.

**Figure 5 molecules-25-01439-f005:**
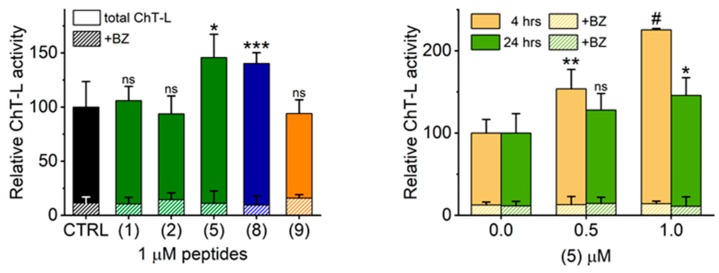
Treatment of human neuroblastoma SK-N-SH (ATCC HTB-11) cell line with selected TAT1 derivatives resulted in significant augmentation of the proteasome activity. Left: After 24 h treatment with vehicle (DMSO) or TAT compound, cell lysates were prepared and ChT-L activity measured. Relative activity per mg of protein in lysates is presented. Treatment of the same cell lysates with 1 µM Bortezomib lead to almost complete abrogation of ChT-L activity independent of the type of TAT peptide used. Proliferation and viability of cells were not affected by the treatments: live cell counts remained at the level of 95%–107% of control, with the sole exception of a lower count for (9) (86%, non-significant difference). The content of dead cells varied between 9% and 12% for all samples (no significant differences). Right: SK-N-SH cells were treated for four or 24 h with 0.5 or 1.0 µM of (5) and processed as above. An increase of proteasome activity was dose dependent and was affected by the treatment interval. Exposure of the cell lysates to 1 µM BZ almost completely eliminated ChT-L activity and its dependence on treatment time and the applied (5) dose. The control 100% activity corresponded to 1.1 (24 h treatments) or 0.31 (4 h treatments) nanomol of the AMC product released by mg of the lysate per minute. *, *p* ≤ 0.05; **, *p* < 0.01; ***, *p* < 0.001; #, *p* < 0.00001 (t test; *n* = 3–4); ns - not significant difference.

**Figure 6 molecules-25-01439-f006:**
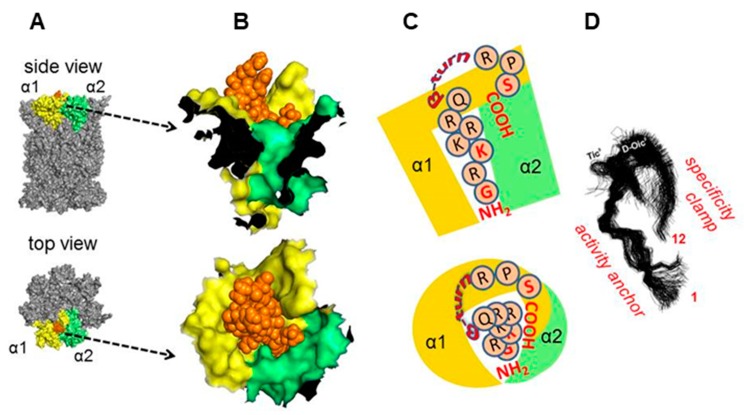
Molecular docking (Rhodium^®^) of (5) to the human 20S (pdb: 5LE5; [35]), pointed to the α1/α2 pocket as the preferred binding site. (**A**) Surface rendering of a fragment of the α face with (5) protruding from the pocket; below–α face with orange colored (5) and α1/α2. (**B**) Zoom-in of proposed positioning of (5) in the binding pocket. (**C**) Schematic positioning of (5) in the α face binding pocket. (**D**) Molecular dynamics simulation of (5) (from [24]) with putative functional significance of the N-terminal and C-terminal fragments for interactions with the proteasome.

**Figure 7 molecules-25-01439-f007:**
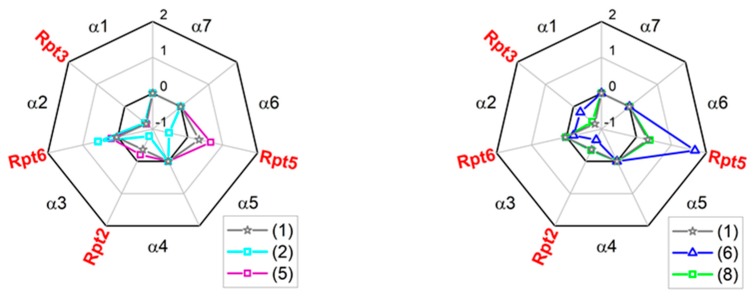
TAT1 derivatives competed with selected C-terminal tails of Rpt (Regulatory Particle ATPases) subunits for binding to specific α face pockets. The scores presented in radar plots were calculated as ((relative activation by the [x] and Rpt tail)-(theoretical sum of activation by [x] and Rpt tail))/(theoretical sum of activation by [x] and Rpt tail), where [x] represent any TAT peptide. Score = 1 indicates a pure additive effect of Rpt and TAT peptides, score < 1 suggests competition, score > 1 hints at synergy. (5) competed fairly specifically with Rpt3 tail, expected to dock in the α1/α2 pocket, as evident from the strongly negative score. (1) and (8) added a weakly negative score for the Rpt2 tail to the presumed major competitor Rpt3. In turn, Rpt2 tail was a major competitor for (6). The poor activator (2) was the least specific ligand of the α face, competing with Rpt2, Rpt3, and Rpt5 tails. The positive scores for combinations of selected compounds with Rpt5 or Rpt6 tails indicated a presumed synergy in activation of the core. Average scores derived from three to six experiments are presented. The competition effects were statistically significant (at least *p* < 0.05) for Rpt3 peptide and (1) and (8), as well as for Rpt2 peptide and (6).

**Figure 8 molecules-25-01439-f008:**
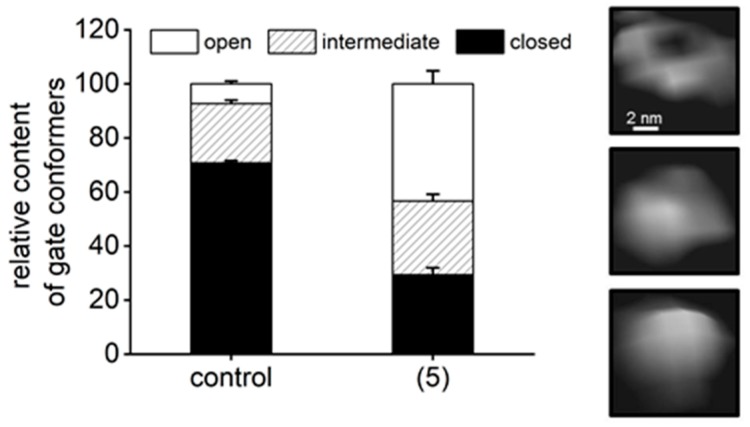
AFM (Atomic Force Microscopy) imaging detected a larger abundance of open-gate conformers after treatment of 20S proteasomes with (5). In control samples treated with vehicle (DMSO) the closed-gate conformation prevailed (71%), with only 7% of open-gate particles and 22% assuming intermediate conformation. The partition for latent, untreated proteasomes was undistinguishable from DMSO-treated controls (not shown and [20,39]). In the presence of 1 μM of (5) 43% of core proteasomes assumed the open-gate conformation at any given time of AFM probing. Only 29% molecules remained in closed conformation, and 27% proteasomes were classified as intermediates. In total, 180 control and 337 (5)-treated particles/cases from *n* = 4 independent experiments were analyzed. Based on the results of the chi squared test, we concluded that the partition of 20S proteasome gate conformers is associated with the presence of (5) and significantly different from the partition found in the control (DMSO-treated) proteasome.

**Table 1 molecules-25-01439-t001:** TAT1 derivatives explored in this study. Compounds (7) and (9) constitute negative controls.

	**Compound**	**Structure**	R1 GRKKRRQ R2 RQRRRPS R3 GRK R4 RQ R5 RPS R6 KKK R7 AAA
(1)	TAT1	GRKKRRQRRRPS	
(2)	TAT1-4,5 TO 8,9-TOD	R3-Tic-Oic-R4-Tic-D-Oic-R5	
(3)	TAT1-4,5 TO	R3-Tic-Oic-R2	
(4)	TAT1-8,9 TO	R1-Tic-Oic-R5	
(5)	TAT1-8,9 TOD	R1-Tic-D-Oic-R5	
(6)	3K-TO-3K	R6-TicOic-R6	
(7)	3A-TO-3A	R7-TicOic-R7	
(8)	TAT1-Den	R6 R6-DABA< R6	
(9)	TAT1 A8-10	GRKKRRQ-R7-PS	

**Table 2 molecules-25-01439-t002:** Peptides based on HIV-1 Tat protein fragment, Tat1, are potent activators of the 20S core proteasome in vitro (purified 20S proteasome in Tris/HCl buffer pH 8 with model substrate for the ChT-L activity). AC_50_: concentration at which the compound reaches 50% of the maximal activation effect. Titration curves for the compounds are presented in Figure 3.

	Compound	AC_50_ [nM] ± SE	Max Activation Fold ± SE
(1)	Tat1	710 ± 12	6.2 ± 0.20
(2)	Tat1-4,5 TO 8,9-TOD	1528 ± 208	1.5 ± 0.26
(3)	Tat1-4,5 TO	643 ± 74	3.3 ± 0.23
(4)	Tat1-8,9 TO	499 ± 62	5.2 ± 0.76
(5)	Tat1-8,9 TOD	220 ± 89	7.9 ± 0.74
(6)	3K-TO-3K	1199 ± 118	8.0 ± 0.12
(7)	3A-TO-3A	1744 ± 414	1.2 ± 0.04
(8)	Tat1-Den	374 ± 13	9.8 ± 0.18
(9)	TAT1 A8-10	ND	0.9 ± 0.12

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
