# Peer review of "New Peptide-Based Pharmacophore Activates 20S Proteasome"

_molecules, 2020, doi:10.3390/molecules25061439_

Round 1
Reviewer 1 Report
An interesting study on TAT-related peptides as proteasome stimulators. Only a few things need to be modified. Overall, it is an interesting study on proteasome activation with a new set of peptides.
-The statement about the animal studies in the abstract is confusing, there aren't animal studies in this manuscript.
-What is the serum stability of these peptides? Since they are peptides they will far apart fairly quickly. So I assume that Table 2 and Figure 3 is only done in buffer and purified 20S? I am really surprised that the TAT peptides survive 24 hrs in cell media to make such a significant impact on proteasome activity. Does the serum stability data reflect this?
-It would be of interest for the authors to note why compound 8 and 6 are not bell curves like the other
-More information about the molecular docking is required. This is not the first paper I have seen do this, and it is unclear how programs can handle such large complexes and that this data is actually valid. Surely there were other potential binding site locations, what makes the authors so confident if it the alpha-1/2 interface? It seems very often this is always the surface people calculate that molecules interact with.
Reviewer 2 Report
The Authors investigated the 20S proteasome-activating capacity of HIV TAT protein-derived peptides following the generation of various related peptides by measuring chymotripsin-like peptidase activity in vitro. For predicting the mechanism of action, the Auhors performed in silico modeling and competition experiments using selected TAT compounds after treatment with Rpt peptides. They conclude that some of the peptides possess potential proteasome-activating capacity.
Data presented here has already been published in a U:S. patent. as őer Authors declaration.
Below please find my major criticisms.
The Authors draw conclusions from the invitro peptidase activity experiments comparing the tested peptides without carrying out appropriate statistical analyses. The reasoin is not clear and the Authoors themselves might notice this flaw of the manuscript as they often communicate their results as sugesstions. This way, however, scientific data cannot be communicated in an original paper so the variance analysis of tdata presented needs to be performed before submitting the manuscript. The manuscript identifies peptides with significant proteasome activating capacity but these observations are not followed by in vivo experiments. this represents one of the weakest point of the manuscript. The manuscript needs to be rephrased in multiple partsfor clarification. The manuscript has grammatical errors that needs to be corrected.Author Response
Please see attachment

Reviewer 3 Report
The original research article by Osmulski and colleagues describes the development of new peptide-based compounds that stimulate the activity of the proteasome. The study is based on the design and description of a series of modified peptides derived from the HIV TAT protein, that was identified in a previous study by the authors, to stimulate the proteasome. The study is well conducted and offers valuable insight into the development of proteasome agonists. A few questions/suggestions remain.
Major comments:
Figure 2: The authors should give the P-value, or at least ‘star-coding’ them, for the different constructs. Statistical analysis should be detailed in the figure legends were relevant and in a methods subsection. Figure 5: Would it be possible to check at lower concentrations if the effect of the best peptides is maintained (at least for one of the peptides)? In their in cellulo experiments the authors used human neuroblastoma SK-N-SH lines. Is there any information about the basal activity level of the proteasome is these cells compared to non-cancerous or other cancerous cell lines? It would be interesting to compare the impact of the bioactive peptides in a non-cancerous cell line. Figure 8: The authors have performed several repeats of the experiments, it would be valuable to add the results from the statistical analysis. In the method section, the authors mentioned using bortezomib as a proteasome inhibitor for the cell culture experiments to exclude the activity of other degradative pathways on the degradation of the substrate. Could the authors add the bortezomib data to the appropriate figure(s) as additional controls?
Minor comments:
Line 102: the sentence is oddly truncated. Maybe due to formatting following insertion of the figure? Line 133: susspect -> suspect Line 143: did the authors meant to cite Figure 3 instead of Figure 4? Figure 4 should be cited with the following sentence. Line 323: the sentence starting as “Still, in our in vitro experiment…” is confusing and may need to be rephrased. References 10 and 12 are the same. Maybe the authors could replace one with the following review: Njomen, E., & Tepe, J. J. (2019). Proteasome Activation as a New Therapeutic Approach To Target Proteotoxic Disorders. Journal of medicinal chemistry, 62(14), 6469–6481. doi:10.1021/acs.jmedchem.9b00101Author Response
Please see attachment

Round 2
Reviewer 2 Report
Authors addressed issues.
Author Response
We are very pleased that the Reviewer found our response satisfactory.
Reviewer 3 Report
Thank you to the authors for their response. All my comments were taken into account and satisfactory answers and/or modifications to the manuscript were made.
Minor comment:
To fully accept the manuscript, it would be beneficial if the authors could add the information related to the statistical analysis in the method section (tests ran, software).
Author Response
We are very grateful for the suggestion. Indeed, a separate "Statistical analysis" description has been missing. We added the respective paragraph, copied below:
4.7 Statistical analysis
All experiments were performed at least in three independent replicates. The results are presented as a mean ± SD. A two-tailed t-test was applied to compare the means. The comparisons were allowing for unequal variances with a Welch correction. Abundance of proteasome conformers was compared with the chi squared test and two-sample proportion test. A significance level was set at 0.05. Statistical analysis was performed using statistical procedures offered by OriginPro 2019 (OriginLabs, Northampton, MA). Reaction rates were calculated from a smoothed linear segment of kinetic traces using OriginPro 2019. Response curves (activity vs compound concentration) were fitted with a nonlinear fitting application of OriginPro 2019. The AC50 and maximum activation fold was calculated based on the equations implemented to fit the response curves.